# Advancing Table Understanding of Large Language Models via Feature Re-ordering

## Abstract

Large Language Models (LLMs) exhibit exceptional proficiency in comprehending human language. Despite their significant success across a wide array of tasks, including text generation, translation, question answering, and even code generation, understanding tabular data remains a challenging task. Especially, tabular data lacks an intrinsic order of the different features (table fields), whereas LLMs take only sequential inputs. Consequently, an artificial order is imposed, the impact of which on the performance of LLMs has not yet been thoroughly investigated. Surprisingly, as discovered in this work, this artificially induced order bias dramatically influences the performance of LLMs on tasks related to tabular data. Mitigating the order bias presents a significant challenge. To address this, we propose a simple and cost-effective method, Re-Ordering Tabular feATures fOR LLM (ROTATOR-LLM), to conduct test-time compute without fine-tuning the base LLM. Aiming at optimizing the feature order of tabular data and boosting LLMs' capability to better understand the data semantics, ROTATOR-LLM re-frames the ordering problem as a feature trajectory generation task. A dynamic programming based meta-controller is trained to auto-regressively generate an individualized feature trajectory for each data instance via accumulative value estimation of the serialized feature input through the LLM's final performance metrics. Model performance is maximized by iteratively selecting features across different steps. Experimental results on multiple datasets and LLMs show close to or over 20% performance boosts via features reordered by ROTATOR-LLM against the un-ordered counterpart. Also, it outperforms State-Of-The-Art tabular LLM methods with significant margin. Moreover, meta-controller demonstrates strong transferability: the tested LLMs gain performance enhancements when utilizing a meta-controller trained on one of them.

## 1 Introduction

Tabular data is prevalent in real-world scientific, medical, biological, sociological, financial, and retail databases, necessitating significant time and effort for humans to process and analyze Dong & Wang (2024); Fang et al. (2024). Fortunately, advancements in large language models (LLMs) have enabled rigorous exploration of their application in various tasks related to tabular data modeling Yuan et al. (2024); Hu et al. (2024). Recent breakthroughs have involved LLMs to handle a wide range of tabular data tasks, such as TabLLM Hegselmann et al. (2023), TableGPT Zha et al. (2023b), and TableLlama Zhang et al. (2023).

Although tabular data can be easily converted into text format, LLMs struggle to effectively analyze the converted data. Since LLMs are primarily pre-trained on natural language, they face challenges in extracting meaningful insights from structured tabular data. To overcome this challenge, existing work primarily focuses on fine-tuning LLMs on tabular dataset to inject the data prior knowledge to the models. For example, TableLlama employs LongLoRA to fine-tune the Llama-2-7B LLM on the extensive TableInstruct datasets. Similarly, TableGPT introduces a table encoder and chain-of-command mechanism, utilizing a Phoenix-7B LLM for inference. Despite these advancements, much of the current research on tabular data analysis overlooks the critical role of feature order in the prompt: due to the sequential nature of transformer decoder based models, an artificial order is inevitably created when feeding the features into the LLM one by one regardless of the detailed prompting schemes. Our extensive studies reveal that this induced ordering of features significantly

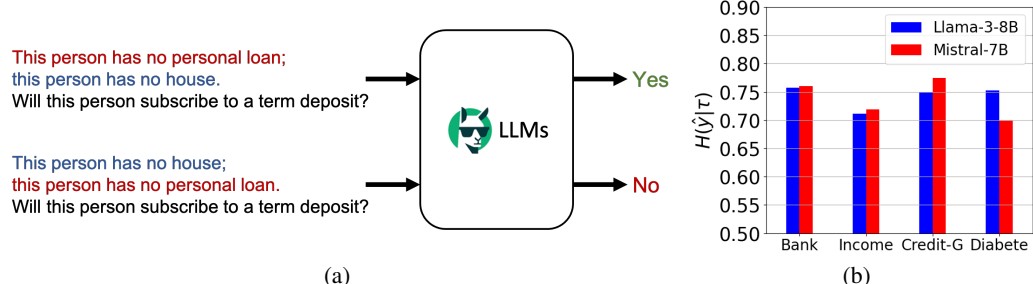

(a)                                                         (b)

Figure 1: (a) An example of LLM order bias. (b) Order bias generally exist in different LLMs.

impacts LLM's behavior Chen et al. (2024); Xu et al. (2024). For instance, the LLM prediction on the same data instance can vary just by changing the order of input features, as in Figure 1 (a). Further details are discussed in Section 3.

This problem is mainly rooted in the order bias in the pre-training data, where the collected data follows certain sequences preferred by humans. Such order preference is captured by the LLMs during the pre-training stage, which enables LLMs to better learn the data semantics whose feature importance ranking aligns with the order bias Sagawa et al. (2019); Koh et al. (2021). To tackle this, an intuitive solution is to remove the order bias by fine-tuning the LLMs on unbiased data. However, fine-tuning LLMs is not only time- and resource-consuming due to the billions of updated parameters, but also labor-intensive, requiring collecting high-quality data Yang et al. (2024); Zha et al. (2023a). A more practical approach is to preprocess the data to align with the LLMs' inherent order bias, enabling them to better grasp the data's semantics. This alignment offers greater potential for real-world applications due to its feasibility, scalability, and extensibility across diverse datasets.

In this work, we introduce Re-Ordering Tabular feATures fOR LLM (ROTATOR-LLM), a simple and cost-effective method to help LLMs better comprehend data semantics via test-time compute in the input level Snell et al. (2024). Specifically, ROTATOR-LLM converts the feature ordering problem into a task of generating feature trajectories, where each trajectory represents a sequence of features in a specific order. To avoid the high resource consumption of fine-tuing the LLM and the corresponding expensive human labeling, ROTATOR-LLM trains a light-weight neural network as a meta-controller to auto-regressively generates the optimized feature trajectory for each data instance, guided by a value function designed to supervise its training process. It is challenging to define the value function for a specific feature order such that this value aligns with the corresponding LLMs' performance. We are motivated by dynamic programming to overcome this challenge. Specifically, the value of a feature trajectory is defined as its potential maximal value in the next state within the whole generation path. At the last state, the value of an integral trajectory is determined by the LLMs' performance. This approach allows us to estimate the value of any feature trajectory, which, in turn, supervises the training of the meta-controller. To evaluate ROTATOR-LLM, we conduct experiments with three LLMs across four tabular datasets. The results demonstrate that LLMs perform significantly better on data reordered by ROTATOR-LLM compared to random or default orders, underscoring the effectiveness of the reordering process. Moreover, ROTATOR-LLM outperforms existing foundational tabular LLMs, further highlighting its potential in real-world applications. In summary, our contributions in this work are as follows:

- **Order Bias of LLMs.** We demonstrate that the order of instance features in a prompt significantly influences LLM predictions, identifying the presence of order bias.

- **Alignment to Order Bias.** We propose ROTATOR-LLM, a cost-effective solution that requires no tuning of LLM parameters. ROTATOR-LLM aligns a data instance with the inherent order bias of LLMs by re-ordering its features.

- **Experimental Evaluation.** Experimental results on four datasets with three popular LLMs demonstrate the superior performance lift brought by ROTATOR-LLM, which improves LLMs' classification accuracy by 20% in average.

| Last-layer Attention Map | | | |
|---|---|---|---|
| ~ 1 | | | |
| ~ 2,1 | ~ 2,1 | | |
| ~ 3,2,1 | ~ 3,2,1 | ~ 3,2,1 | |
| ~ 4,3,2,1 | ~ 4,3,2,1 | ~ 4,3,2,1 | ~ 4,3,2,1 |

Feature 1 → LLM
Feature 2 →
Feature 3 →
Feature 4 →

(a) Attention matrix of features 1, 2, 3, and 4.

| Last-layer Attention Map | | | |
|---|---|---|---|
| ~ 2 | | | |
| ~ 3,2 | ~ 3,2 | | |
| ~ 4,3,2 | ~ 4,3,2 | ~ 4,3,2 | |
| ~ 1,4,3,2 | ~ 1,4,3,2 | ~ 1,4,3,2 | ~ 1,4,3,2 |

Feature 2 → LLM
Feature 3 →
Feature 4 →
Feature 1 →

(b) Attention matrix of features 2, 3, 4, and 1.

Figure 2: Comparison of the last-layer attention map under different orders of input features. Since each feature is represented by a sentence, i.e. multiple tokens, each cell corresponds to a matrix of attention values between tokens. The notation '$\sim i, j, k$' indicates the attention matrix is computed based on a mixture of information from the token embeddings associated with features $i, j$ and $k$.

## 2 PRELIMINARIES

We introduce the notations and data format transition in this section.

### 2.1 NOTATIONS

We consider aligning the dataset $\mathcal{D} = (\boldsymbol{x}, y) \mid \boldsymbol{x} \in \mathcal{X}, y \in \mathcal{Y}$ to the order bias of LLMs $f(\bullet)$. Each instance $\boldsymbol{x} \in \mathcal{X}$ has $M$ features, $\boldsymbol{x} = [x_1, x_2, \cdots, x_j, \cdots, x_M]$, where $j \in \mathcal{J} = \{1, 2, \cdots, M\}$ is the feature index in the default order of a particular tabular dataset. Let $\boldsymbol{\tau} = [\tau_1, \tau_2, \cdots, \tau_M]$ denote a specific ordering of the features of instance $\boldsymbol{x}$, representing a feature trajectory with $M$ positions. For $1 \leq t \leq M$, each $\tau_t \in \{x_1, x_2, \cdots, x_M\}$ indicates a feature ranked at position $t$; and $\boldsymbol{\tau}_{[0:t]}$ denotes a slice of the trajectory comprising the first $t$ positions $[\tau_1, \cdots, \tau_t]$, each containing a feature best suited for the corresponding position. The case $t = 0$ represents the initial state $\boldsymbol{\tau}_{[0:0]} = [\ ]$ where no features have been ranked, while $t = M$ denotes the final state $\boldsymbol{\tau}_{[0:M]}$ that all $M$ positions are filled by properly ranked features. For example, if there are in total 3 features, the full trajectory $\boldsymbol{\tau} = [x_2, x_3, x_1]$ represents the features are ordered as 2, 3, and 1 at positions 0, 1, and 2, respectively. In Section 3, we demonstrate the order bias of LLMs by showing that the prediction results $\hat{y} = f(\boldsymbol{\tau})$ are significantly affected by the order of input features $\boldsymbol{\tau}$. To address this issue, we introduce ROTATOR-LLM in Section Section 4, which aligns the dataset $\mathcal{D}$ with the order bias of LLMs. ROTATOR-LLM aims to generate the optimal trajectory $\boldsymbol{\tau}^*$ for each instance $\boldsymbol{x}$, thereby maximizing the accuracy of the LLMs' predictions.

### 2.2 TEXT-BASED SERIALIZATION

Text-based Serialization refers to converting tabular data into text data to fit the input modality of LLMs. Existing work explores several methods of text-based serialization. For example, Markdown table Liu et al. (2023); Jaitly et al. (2023), JSON-file format Singha et al. (2023); Sui et al. (2024), and sentence serialization Yu et al. (2023); Jaitly et al. (2023). To maximally leverage the sequence-to-sequence capacity of LLMs, we consider the sentence serialization to convert the data features into text data. The advantage of sentence serialization is its alignment with the natural language data where LLMs are pre-trained. In this work, we use a template given in Appendix A to convert tabular data into text data. For instance, we adopt the sentence "the age of this person is 30; this person has no house" to represent the tabular data {`Age:30,House:No`}. Our method can be easily extended to fit Markdown table and JSON-file formats of serialized data, but their performance is out of the scope of this work.

## 3 ORDER BIAS OF LLMS ON TABULAR DATA

In this section, we empirically analyze the order bias of LLMs and present the experimental evidence of LLM's behavior change under the influence of order bias.

### 3.1 WHY LLMs HAVE ORDER BIAS?

Order bias refers to the impact that the sequence of tabular data features has on the predictions made by LLMs. While from the perspective of how human beings understand the tabular data, the order of features/fields is not meaningful and should not affect the model output, each particular serialization of these features/fields indeed results in a different input sequence for an auto-regressive model and accordingly a difference in the outcome. For LLMs, this difference affects their attention maps. We show an example in Figure 2 to demonstrate the influence of different feature orders on the last-layer attention maps. As each feature is represented by a sentence, i.e. multiple tokens, each cell in Figure 2 corresponds to a matrix of attention values between tokens. The notation '$\sim i, j, k$' indicates the attention matrix is computed based on a mixture of information from the token embeddings associated with features $i, j$ and $k$. In this example, the sequence of features 1, 2, 3, and 4 in Figure 2 (a) mixes a different set of tokens compared to the feature sequence of 2, 3, 4, and 1 for the computation of last-layer attention map. The variations in last-layer attention maps lead to obvious differences in the prediction results.

### 3.2 DEMONSTRATIONS OF ORDER BIAS

We demonstrate the presence of order bias in LLMs using real-world tabular datasets. Specifically, we examine the variance in LLMs' predictions caused by different permutations of data features. The probability of LLMs' predictions is estimated by $\mathbb{P}(\hat{y} = 1) = \frac{\# \text{ of } 1}{\# \text{ of Permutations}} = \frac{\# \text{ of } 1}{M!}$, and $\mathbb{P}(\hat{y} = 0) = 1 - \mathbb{P}(\hat{y} = 1)$. The variance in predictions is quantified by the entropy $\mathcal{H}(\hat{y}) = -\mathbb{P}(\hat{y} = 0)\log_2 \mathbb{P}(\hat{y} = 0) - \mathbb{P}(\hat{y} = 1)\log_2 \mathbb{P}(\hat{y} = 1)$. For instance, for data instance having two features: age and house, if an LLM outputs $\hat{y} = 1$ for {`Age:30,House:No`} and $\hat{y} = 0$ for {`House:No,Age:30`}, then $\mathbb{P}(\hat{y} = 1) = \mathbb{P}(\hat{y} = 0) = 0.5$, resulting in an entropy of 1. If the LLM's predictions show no variance, then either $\mathbb{P}(\hat{y} = 0) = 1$ or $\mathbb{P}(\hat{y} = 1) = 1$, yielding a minimal entropy of 0. Conversely, if the predictions are randomly distributed, $\mathbb{P}(\hat{y} = 0) = 0.5$ and $\mathbb{P}(\hat{y} = 1) = 0.5$, leading to a maximum entropy of 1. Higher entropy indicates greater variance in prediction results, signifying a stronger presence of order bias in the LLMs.

The experiments are conducted on the Bank, Income, German Credit, and Diabete datasets Asuncion et al. (2007), using the `Llama-2-8B-instruct` Touvron et al. (2023) and `Mistral-7B-Instruct` Jiang et al. (2024) LLMs as predictors. The entropy of predictions resulting from feature reordering is shown in Figure 1 (b). Notably, all LLMs applied to the tabular datasets exhibit an entropy exceeding 0.7, approaching the maximum value of 1. This clearly indicates the presence of order bias.

## 4 RE-ORDERING TABULAR FEATURES FOR LLM (ROTATOR-LLM)

In this section, we introduce Re-Ordering Tabular feATures fOR LLM (ROTATOR-LLM) in details. Specifically, ROTATOR-LLM adopts a meta-controller to generate the reordered feature trajectory; then converts the features to text data following the template in Appendix A; finally inputs the data features in text format to LLMs for inference. The overall objective is to maximize the accuracy of the LLM predictions for tabular data classification tasks. We discuss the details as follows.

### 4.1 FEATURE TRAJECTORY GENERATION

ROTATOR-LLM maintains a meta-controller $g(\bullet \mid \theta) : \mathcal{T} \to \mathbb{R}$ to estimate the ranking value of each feature at each location. Specifically, for $0 \le t \le M$, with a slice of trajectory $\boldsymbol{\tau}_{[0:t]}$ as input, the value of $g([\boldsymbol{\tau}_{[0:t]}, x_j] \mid \theta) \in \mathbb{R}$ represents the value of trajectory $[\boldsymbol{\tau}_{[0:t]}, x_j]$, which also indicates the ranking value of feature $j$ at position $t$, given the feature ordering of first $t$ positions $\boldsymbol{\tau}_{[0:t]}$. We consider a higher value $g(\boldsymbol{\tau} \mid \theta)$ as indicative of better ranking results for feature orders that align more closely with the preferences of the LLMs. Therefore, ROTATOR-LLM can recursively generate a trajectory of $M$ data features by

$$\tau_t = \arg\max_{j \in \mathcal{J}} g([\boldsymbol{\tau}_{[0:t-1]}, x_j] \mid \theta). \tag{1}$$

We define a value function $v(\boldsymbol{\tau})$ to compute the classification loss of LLMs' prediction over input data crafted with the feature trajectory $\boldsymbol{\tau}$. We believe a feature ordering that is more aligned with

---

**Algorithm 1** Re-Ordering Tabular feATures fOR LLM (ROTATOR-LLM)

---

**Input:** Training dataset $\mathcal{D}$ and LLM $f(\bullet)$.
**Output:** Meta-controller $g(\bullet \mid \theta)$.

1: **for** $(\boldsymbol{x}, y) \sim \mathcal{D}$ **do**
2:      Generate trajectory $\boldsymbol{\tau}$ by Equation (1) based on initial value $\boldsymbol{\tau}_{[0:0]} = [\,]$.
3:      Estimate the loss value of LLMs' prediction $L_f(f(\boldsymbol{\tau}), y)$.
4:      Estimate the value function $v(\boldsymbol{\tau}_{[0:t]})$ for $1 \le t \le M$ based on Equation (6).
5:      Update the parameters of $g(\bullet \mid \theta)$ to minimize Equation (5).
6: **end for**

---

LLMs' pre-training can lead to better prediction result. Therefore, $v(\boldsymbol{\tau})$ is defined as follows:

$$v(\boldsymbol{\tau}) = -L_f(f(\boldsymbol{\tau}), y) \tag{2}$$

where $L_f$ denotes the cross-entropy; $f(\boldsymbol{\tau})$ is the prediction output of the base LLM; trajectory value function $v(\boldsymbol{\tau})$ is opposite to the cross-entropy loss such that the optimal trajectory $\boldsymbol{\tau}^*$ can minimize the classification error while maximizing the corresponding value function.

Note that Equation (2) only defines the value of a complete trajectory $v(\boldsymbol{\tau})$, it is important to extend its definition to a slice of trajectory $v(\boldsymbol{\tau}_{[0:t]})$, for the purpose of training the controller $g(\bullet \mid \theta)$. However, the value function is strictly defined on the full trajectory $\boldsymbol{\tau}$ (not on its slices) and the final LLM output after feeding $\boldsymbol{\tau}$ into it, so that $v(\boldsymbol{\tau}_{[0:t]})$ cannot be directly obtained via Equation (2). To overcome this challenge, we employ dynamic programming to define $v(\boldsymbol{\tau}_{[0:t]})$, where $0 \le t < M$. Specifically, for a slice of trajectory $\boldsymbol{\tau}_{[0:t]}$, its value function $v(\boldsymbol{\tau}_{[0:t]})$ is defined as the maximal value of $v(\tilde{\boldsymbol{\tau}})$ such that $\tilde{\boldsymbol{\tau}}_{[0:t]} = \boldsymbol{\tau}_{[0:t]}$, which is given by

$$v(\boldsymbol{\tau}_{[0:t]}) = \max_{\tilde{\boldsymbol{\tau}}_{[t-1:M]}} \gamma^{M-t} v([\boldsymbol{\tau}_{[0:t-1]}, \tilde{\boldsymbol{\tau}}_{[t-1:M]}]), \tag{3}$$

$$= \max_{j \in \mathcal{J}} \gamma v([\boldsymbol{\tau}_{[0:t-1]}, x_j]), \tag{4}$$

where $0 < \gamma < 1$ denotes a discounting factor. The discounting factor regulates how features ranked at different positions cumulatively contribute to the final cross entropy and full trajectory value. This is inspired by the observation in previous studies that tokens closer to the end contribute relatively more to the output of LLMs Jin et al. (2024).

According to Equation (4), we have an iterative property of the value function given by $v(\boldsymbol{\tau}_{[0:t]}) = \gamma v(\boldsymbol{\tau}_{[0:t+1]})$ running backwards from positions $t = M$ to $t = 0$ with the last state value given by $v(\boldsymbol{\tau}) = -L_f(f(\boldsymbol{\tau}), y)$ at $t = M$. Therefore, the parameters of $g(\boldsymbol{\tau}_{[0:t]} \mid \theta)$ is updated to minimize the mean-square error aligned with the value function $v(\boldsymbol{\tau}_{[0:t]})$ as follows:

$$L_\theta = \frac{1}{M} \sum_{t=0}^{M} \left[ g(\boldsymbol{\tau}_{[0:t]} \mid \theta) - v(\boldsymbol{\tau}_{[0:t]}) \right]^2, \tag{5}$$

where $v(\boldsymbol{\tau}_{[0:t]})$ can be estimated based on its iterative property as follows:

$$v(\boldsymbol{\tau}_{[0:t]}) = \begin{cases} \gamma \max_j g([\boldsymbol{\tau}_{[0:t]}, x_j] \mid \theta) & \text{if } t < M, \\ -L_f(f(\boldsymbol{\tau}), y) & \text{if } t = M. \end{cases} \tag{6}$$

### 4.2 ALGORITHM OF ROTATOR-LLM

Algorithm 1 shows one epoch of ROTATOR-LLM. Specifically, for each mini-batch of instances, ROTATOR-LLM first generate an order of features following Equation (1) (line 2); then estimate the loss function of LLMs' prediction, where the input data of LLMs follows the generated feature order (line 3); then estimate the value functions based on Equation (6) (line 4); finally updates the parameters of meta-controller to minimize the loss function given in Equation (5) (line 5).

## 5 EXPERIMENTS

In this section, we conduct experiments to evaluate ROTATOR-LLM, aiming to answer the following research questions: **RQ1:** Does ROTATOR-LLM effectively align the data with the LLMs for better

Table 1: Balance accuracy of ROTATOR-LLM on the Bank, Income, Germen Credit, and Diabetes datasets.

| Datasets | Order | Bank | Income | Germen Credit | Diabetes | Average |
|---|---|---|---|---|---|---|
| Llama-3-8B | Default | 0.522 | 0.516 | 0.521 | 0.312 | 0.468 |
|  | Random | 0.510 | 0.520 | 0.535 | 0.385 | 0.488 |
|  | ROTATOR-LLM | **0.791** | **0.752** | **0.665** | **0.738** | **0.737** |
| Mistral-7B | Default | 0.599 | 0.540 | 0.493 | 0.699 | 0.585 |
|  | Random | 0.574 | 0.577 | 0.546 | 0.676 | 0.593 |
|  | ROTATOR-LLM | **0.782** | **0.801** | **0.701** | **0.722** | **0.752** |
| Phi-3-mini | Default | 0.504 | 0.510 | 0.405 | 0.634 | 0.513 |
|  | Random | 0.481 | 0.521 | 0.440 | 0.655 | 0.524 |
|  | ROTATOR-LLM | **0.712** | **0.771** | **0.665** | **0.743** | **0.723** |

Table 2: F1 score of ROTATOR-LLM on the Bank, Income, Germen Credit, and Diabetes datasets.

| Datasets | Order | Bank | Income | Germen Credit | Diabetes | Average |
|---|---|---|---|---|---|---|
| Llama-3-8B | Default | 0.466 | 0.674 | 0.600 | 0.191 | 0.483 |
|  | Random | 0.555 | 0.676 | 0.605 | 0.353 | 0.547 |
|  | ROTATOR-LLM | **0.811** | **0.796** | **0.732** | **0.774** | **0.778** |
| Mistral-7B | Default | 0.428 | 0.678 | 0.145 | 0.691 | 0.486 |
|  | Random | 0.456 | 0.692 | 0.365 | 0.695 | 0.552 |
|  | ROTATOR-LLM | **0.774** | **0.808** | **0.734** | **0.765** | **0.770** |
| Phi-3-mini | Default | 0.245 | 0.664 | 0.182 | 0.505 | 0.399 |
|  | Random | 0.439 | 0.660 | 0.512 | 0.632 | 0.561 |
|  | ROTATOR-LLM | **0.658** | **0.776** | **0.622** | **0.763** | **0.705** |

performance? **RQ2:** Can the controller be transferred between different LLMs? **RQ3:** How does the reordering intrinsically impact the LLMs?

## 5.1 EXPERIMENT SETUP

We specify the datasets, LLMs, baseline methods, evaluation metrics, and implementation details.

**Datasets.** The evaluation of ROTATOR-LLM is based on the Bank, Income, German Credit, and Diabetes datasets from the areas of social media, finance and healthcare. The datasets source from the UC Irvine machine learning repository Asuncion et al. (2007). On each dataset, the data features are first reordered; then converted into text data following the template in Appendix A; and finally being input to LLMs for classification.

**LLMs.** We evaluate ROTATOR-LLM using three popular model families: Llama-3-8B Touvron et al. (2023), Mistral-7B Jiang et al. (2024), and Phi-3-mini-4k Li et al. (2023). These LLMs are employed due to their leadership among open-sourced LLMs according to existing leaderboards Chiang et al. (2024). We download their instruct-tuned version from the Huggingface Wolf et al. (2019).

**Baseine Methods.** We consider four baseline methods compared with ROTATOR-LLM. **Default order.** The features of each data instance follow the default order provided by the datasets. **Random order.** The features of each data instance are randomly ordered. **TableLlama.** A Llama-based foundational tabular LLM fine-tuned on large-scale tabular datasets Zhang et al. (2023). **TableLLM.** A GPT-2-based foundational tabular LLM fine-tuned on large-scale tabular datasets Zha et al. (2023b).

**Evaluation Metrics.** Due to the imbalance of positive and negative examples in the datasets, the regular accuracy metric is not sufficient to truly reflect the classification performance. Therefore, we evaluate the balance accuracy ($\uparrow$) and F1 score ($\uparrow$) of LLMs' classification on the datasets. To estimate the balance accuracy, the instances of the minority class are first duplicated to align with the size of the majority class. Then the accuracy is calculated.

Table 3: Transfer-ability of ROTATOR-LLM, where the meta controller is trained with a source LLM and tested on a different target LLM.

| Metric | Configuration | Bank | Income | Germen Credit | Diabetes | Average |
|---|---|---|---|---|---|---|
| Balance accuracy | Default-Llama | 0.522 | 0.516 | 0.521 | 0.312 | 0.468 |
| | Random-Llama | 0.510 | 0.520 | 0.535 | 0.385 | 0.488 |
| | Mistral→Llama | **0.544** | **0.622** | **0.627** | **0.670** | **0.616** |
| | Default-Mistral | **0.599** | 0.540 | 0.500 | 0.699 | 0.585 |
| | Random-Mistral | 0.574 | 0.577 | 0.546 | 0.676 | 0.593 |
| | Llama→Mistral | 0.581 | **0.756** | **0.581** | **0.756** | **0.669** |
| F1 score | Default-Llama | 0.466 | 0.674 | 0.600 | 0.191 | 0.483 |
| | Random-Llama | 0.555 | 0.676 | 0.605 | 0.353 | 0.547 |
| | Mistral→Llama | **0.598** | **0.714** | **0.675** | **0.722** | **0.677** |
| | Default-Mistral | 0.428 | 0.678 | 0.145 | 0.691 | 0.486 |
| | Random-Mistral | 0.456 | 0.692 | 0.365 | **0.695** | 0.552 |
| | Llama→Mistral | **0.504** | **0.743** | **0.414** | 0.690 | **0.588** |

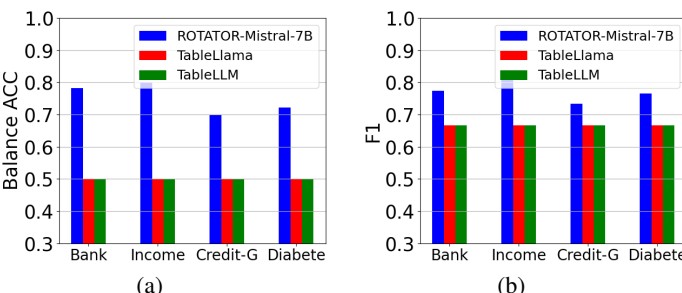

(a)                                                  (b)

Figure 3: Comparison of ROTATOR-LLM with state-of-the-art foundational Table LLMs.

**Implementation Details.** The meta-controller takes a three-layer MLP that is trained using Adam optimizer with learning rate $10^{-3}$ for 200 epochs. An early stop is implemented on the validation datasets. The training and evaluation processes follow the same template of text serialization given in Appendix A. The detailed hyper-parameter setting of ROTATOR-LLM is given in Apendix B.

## 5.2 ALIGNMENT PERFORMANCE (RQ1)

We evaluate the performance of ROTATOR-LLM by examining the classification of LLMs after the alignment. For fair comparison, ROTATOR-LLM and baseline methods adopt the same prompt given in Appendix A for text serialization. The balanced accuracy and F1 score are shown in Tables 1 and 2, respectively. The comparison with baseline foundational tabular LLMs is illustrated in Figure 3. According to the experimental results, we have the following observations:

- **Effectiveness of Alignment.** LLMs show much better performance based on ROTATOR-LLM than the data with default and random feature orders. This indicates that ROTATOR-LLM effectively align the data feature to LLMs, and thereafter enhances LLMs' understanding on the tabular data by optimally reordering the features.

- **Competitive Performance.** ROTATOR-LLM outperforms foundational tabular LLMs, e.g., TableLLM and TableLlama. Compare to these costly fine-tuning methods, ROTATOR-LLM not only saves resources effectively but also shows performance superiority.

- **Consistent Performance.** ROTATOR-LLM is consistently competitive over baseline methods across various LLMs and tabular datasets, indicating its stability and generalizability for real-world applications.

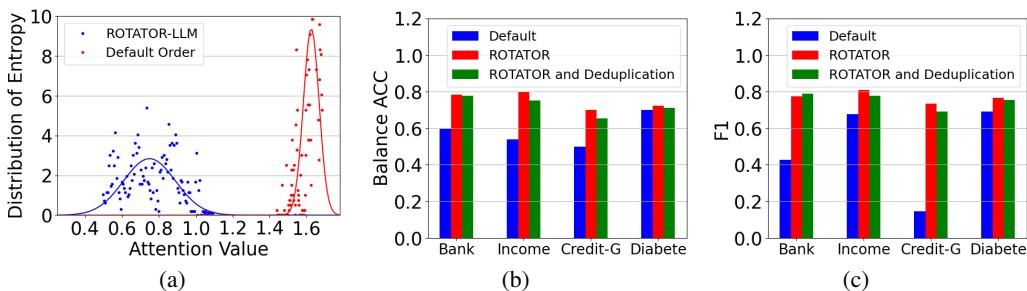

Figure 4: (a) Entropy of last layer attention. The lower the entropy, the more focus of attention. (b) Balanced accuracy and (c) F1 score of shrinking the duplicated features in the prompts.

---

**Prompts:** You are a data analyst. Given information of a person, you should predict whether this person will subscribe to a term deposit. <Data Features> Will this person subscribe to a term deposit?\n\n[Your Response Format]: "Yes / No"

**Label: Yes**

---

**Default features:** This person's age is 33.0. The type of this person's job is technician. This person's marital status is single. This person's education is secondary. This person has no credit in default. This person's average yearly balance in euros is 2979.0. This person has no house. This person has no personal loan. This person's contact communication type is cellular. This person's last contact day of the month is 5.0. This person's last contact month of year is aug. This person's last contact duration is 326.0 seconds. This person has 2.0 contacts performed during this campaign. 437.0 days have passed since this person was last contacted from a previous campaign. This person has 1.0 contacts performed before this campaign. The outcome of this person's previous marketing campaign is failure.

**LLM prediction: No**

---

**Reordered features:** This person's last contact month of year is aug. This person's last contact month of year is aug. This person's last contact month of year is aug. 437.0 days have passed since this person was last contacted from a previous campaign. This person has 1.0 contacts performed before this campaign. The type of this person's job is technician. The type of this person's job is technician. This person has no personal loan. This person's average yearly balance in euros is 2979.0. This person's last contact day of the month is 5. This person has no personal loan. This person's age is 33. This person has no house. This person has no house. The outcome of this person's previous marketing campaign is failure. This person has no personal loan.

**LLM prediction: Yes**

---

**Reorder and Deduplication:** This person's last contact month of year is aug. 437.0 days have passed since this person was last contacted from a previous campaign. This person has 1.0 contacts performed before this campaign. The type of this person's job is technician. This person has no personal loan. This person's average yearly balance in euros is 2979.0. This person's last contact day of the month is 5.0. This person has no personal loan. This person's age is 33.0. This person has no house. The outcome of this person's previous marketing campaign is failure. This person has no personal loan.

**LLM prediction: Yes**

---

Figure 5: Examples of LLM's predictions based on default ordered features, reordered features, and reordered and deduplicated features.

## 5.3 Transfer-ability of Controller (RQ2)

In this section, we evaluate the transferability of the learned controller. The meta-controller is trained based on a source LLM and tested on a target LLM, marked as "source LLM→target LLM". We take Llama-2-8B, Mistral-7B for the source LLMs, and Mistral-7B, Llama-2-8B for the target LLMs, respectively. The results of the controller transfer are shown in Table 3. It is observed that transferring the controller from one LLM to another achieves better performance than inputting the data instance following the default or random order. The results validate the transferability of our learned con-

troller, which meets our expectations as different LLMs could have similar order bias due to the fact that they all focus on learning the large human-generated content in pre-training.

## 5.4 ATTENTION CONCENTRATION BY FEATURE RE-ORDERING (RQ3)

It has been widely shown in existing work Xiao et al. (2023); Zhang et al. (2024b) that the attention of LLM-generated tokens should focus on some key input tokens. Uniform patterns of attention can potentially lead to hallucinations. We conducted experiments to evaluate ROTATOR-LLM in terms of attention concentration. Specifically, this experiment is with Llama-3-8B on the bank dataset using the prompts in Appendix A. The attention is estimated by $\text{softmax}(\mathbf{Q}[:,-1]\mathbf{K}^T/\sqrt{d})$, where $\mathbf{Q}$, $\mathbf{K}$ take the last-layer activations; $d$ takes the hidden dimension value; and the index -1 of $\mathbf{Q}$ indicates the attention is estimated for the answer token. To study the concentration of attention, we show the entropy of last layer attention in Figure 4 (a). The entropy is calculated by $-\sum_{p_j} p_j \log p_j$, where $p_j \sim \text{softmax}(\mathbf{Q}[:,-1]\mathbf{K}^T/\sqrt{d})$ are the attention weights obtained from the softmax operation. Lower entropy corresponds to higher concentrations of attention on a small number of input tokens. It is observed that the last layer attention shows lower entropy after the feature re-ordering than utilizing the default order, indicating more focused attentions on the particular input tokens, rather than uniformly sprout to the whole prompt sequence. This contributes to a better aligned results in Tables 1 and 2.

## 5.5 CASE STUDIES (RQ3)

In this section, we show the data features reordered by ROTATOR-LLM. The data features in natural language sentences are shown in Figure 5, where the place holder <Data Features> takes the "Data features", "Reordered features", and "Reorder and Deduplication" below. We further investigate the affect of deduplication to LLMs' performance in Figure 4, where the deduplication removes the duplicated features from the reordered data. Overall, we have the following insights:

- **Significance of Feature Order.** A good feature order benefits LLMs more than a high number of features. The data instance has 16 features, and only 10 features left after reordering. However, LLMs show more accurate predictions based on the reordered data features.
- **Feature Order is Robust to Deduplication.** The features may be duplicated after the reordering because the features are reordered without replacement. As shown in Figure 4, LLMs maintain the performance to high-levels after removing the redundant features from the input context. This indicates the feature order is robust to the deduplication of redundant features.

## 6 RELATED WORK

We discuss related work on tabular data understanding in this section. Existing work that leverages LLMs to process tabular data is primarily viewed from three perspectives: feature serialization, large-scale fine-tuning, and prompt engineering. We give more details as follows.

**Feature Serialization.**   Feature serialization is a simple way to let LLMs understand tabular data. Specifically, a straightforward way would be to directly input a programming-language readable data structure, such as Markdown format Liu et al. (2023); Jaitly et al. (2023), JSON-file format Singha et al. (2023); Sui et al. (2024), HTML format Singha et al. (2023), and Python dictionary Wang et al. (2023). Another way is to convert the tables into natural language sentence using templates based on the column headers and cell values Yu et al. (2023); Jaitly et al. (2023). This method can maximally leverage the sequence-to-sequence capacity of LLMs to understand tabular data.

**Large-scale Fine-tuning.**   Fine-tuning on tabular datasets is a straightforward way to inject the data prior knowledge to LLMs. There are several existing work of fine-tuning. TableLlama adopts LongLoRA to fine-tune the Llama-2-7B LLM on the extensive TableInstruct datasets Zhang et al. (2023). TableGPT introduces a table encoder and chain-of-command mechanism and performs instruction tunings for Phoenix-7B LLMs on collections of tabular datasets Li et al. (2024). Different from existing work, TabLLM considers few-shot examples for prompts during the fine-tuning, and updates the Bigscience/T0-3B LLMs on single domain tabular datasets Zhang et al. (2024a).

**In-context Learning.** Existing work has demonstrated that LLMs are few-shot learners of tabular data Chen (2022); Narayan et al. (2022); Guo et al. (2023). Leveraging few-shot examples in the prompts, LLMs can better understand the data semantics through in-context learning. Other prompt engineering methods include chain-of-thoughts Wei et al. (2022), tree-of-thoughts Yao et al. (2024), self-consistency Wang et al. (2022), and others Sui et al. (2023).

# 7 CONCLUSION

In this work, we demonstrate novelly discover and thoroughly explore the order bias of LLMs on tabular data, where the arrangement of data features can mislead LLM predictions. To address this issue, we propose ROTATOR-LLM, an approach designed to align tabular data with this order bias, enabling LLMs to better comprehend the data semantics. Specifically, ROTATOR-LLM employs a meta-controller to learn the optimal feature order. It estimates the value function for each feature order using dynamic programming, which guides the training of the meta-controller. Our experimental results on four datasets across three LLMs show that ROTATOR-LLM achieves superior performance compared to baseline methods and state-of-the-art foundational tabular LLMs when applied to reordered data. Additionally, ROTATOR-LLM exhibits strong transferability across multiple LLMs, indicating its adaptability to diverse tasks. Without requiring fine-tuning of LLMs, ROTATOR-LLM proves to be a more cost-effective solution than traditional debiasing methods, underscoring its potential for real-world applications.

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

```
1   table2text_template = {
2       "age": "This person's age is {}.",
3       "job": "The type of this person's job is {}.",
4       "marital": "This person's marital status is {}.",
5       "education": "This person's education is {}.",
6       "default": {"no": "This person has no credit in default.",
7           "yes": "This person has credit in default."},
8       "balance": "This person's average yearly balance in euros is {}.",
9       "housing": {"no": "This person has no house.",
10          "yes": "This person owns houses."},
11      "loan": {"no": "This person has no personal loan.",
12          "yes": "This person has personal loan."},
13      "contact": "This person's contact communication type is {}.",
14      "day": "This person's last contact day of the month is {}.",
15      "month": "This person's last contact month of year is {}.",
16      "duration": "This person's last contact duration is {} seconds.",
17      "campaign": "This person has {} contacts performed during this
            campaign.",
18      "pdays": "{} days have passed since this person was last contacted
            from a previous campaign.",
19      "previous": "This person has {} contacts performed before this
            campaign.",
20      "poutcome": "The outcome of this person's previous marketing campaign
             is {}.'",
21  }
```

Figure 6: Table to Text data template on the bank dataset.

```
1   table2text_template = {
2       "workclass": "The class of this person's job is {}.",
3       "marital_status": "This person's marital status is {}.",
4       "education": "This person's education is {}.",
5       "occupation": "This person's job is {}.",
6       "relationship": "This person's relationship in family is {}.",
7       "sex": "This person's gender is {}.",
8       "race": "This person's race is {}.",
9       "native_country": "The native country of this person is {}.",
10      "age": "This person's age is {}.",
11      "fnlwgt": "The final analysis weight of this person is {}.",
12      "education_num": "The education duration of this person is {}.",
13      "capital_gain": "The capital gain of this person is {}.",
14      "capital_loss": "The capital loss of this person is {}.",
15      "hours_per_week": "The person works {} hours per week in average.",
16  }
```

Figure 7: Table to Text data template on the Income dataset.

# APPENDIX

## A  TEMPLATE OF TEXT-BASED SERIALIZATION

We give the template of text-based serialization in this work. The templates for the bank, Income, German Credit, and Diabete datasets are given in Figures 6, 7, 8, and 9, respectively.

## B  HYPER-PARAMETER SETTING OF ROTATOR-LLM

The hyper-parameter setting of ROTATOR-LLM in Table 4. The discounting factor for meta-controller training is given in Table 5.

```
1  table2text_template = {
2      "checking_status": "The_status_of_this_person's_checking_account_is_
           {}.",
3      "credit_history":  "The_status_of_this_person's_historical_credits_is
           _{}.",
4      "purpose": "This_person's_purpose_to_apply_for_credits_is_{}.",
5      "savings_status": "The_status_of_this_person's_saving_account_is_{}."
           ,
6      "employment": "The_present_employment_of_this_person_is_{}.",
7      "personal_status": "The_marital_status_of_this_person_is_{}.",
8      "other_parties": {"none": "This_person_does_not_have_other_debtors.",
9          "co_applicant": "This_person_has_co-applicants.",
10         "guarantor": "This_person_has_guarantors."} ,
11     "property_magnitude": "The_property_magnitude_of_this_person_is_{}.",
12     "other_payment_plans": {"none": "This_person_does_not_have_other_
           installment_plans.",
13         "stores": "This_person_has_installment_plans_for_stores.",
14         "bank": "This_person_has_installment_plans_for_banks."},
15     "housing": {"own": "This_person_owns_houses.",
16         "rent": "This_person_rents_a_house.",
17         "for_free": "This_person_lives_in_a_free_house."},
18     "job": "The_type_of_this_person's_job_is_{}.",
19     "own_telephone": {"none": "This_person_does_not_have_a_telephone.",
20         "yes": "This_person_owns_a_telephone."},
21     "foreign_worker": {"yes": "This_person_is_a_foreign_worker.",
22         "no": "This_person_is_not_a_foreign_worker."},
23     "duration": "The_duration_of_this_person_is_{}_months.",
24     "credit_amount": "The_amount_of_this_person's_credit_is_{}.",
25     "installment_commitment": "This_person_has_a_installment_rate_of_{}_
           of_disposible_income.",
26     "residence_since": "This_person_has_been_a_residence_for_{}_years.",
27     "age": "This_person's_age_is_{}.",
28     "existing_credits": "This_person_already_has_{}_credits.",
29     "num_dependents": "This_person_supports_{}_dependents.",
30  }
```

Figure 8: Table to Text data template on the Germen Credit dataset.

| Name | Value |
|---|---|
| Layer Number | 3 |
| Hidden Dimension | 512 |
| Optimizer | Adam |
| Learning Rate | 0.001 |
| Epoch | 200 |
| Mini-batch Size | 128 |

Table 4: Hyper-parameter setting of ROTATOR-LLM.

| | Bank | Income | German Credit | Diabete |
|---|---|---|---|---|
| Llama-3-8B-Instruct | 0.75 | 0.8 | 0.8 | 0.8 |
| Mistral-7B-Instruct | 0.85 | 0.9 | 0.85 | 0.9 |
| Phi-3-Mini-Instruct | 0.9 | 0.8 | 0.8 | 0.8 |

Table 5: Discounting factor on meta-controller training.

```
table2text_template = {
    "HighBP": {0: "This person has a normal blood pressure.",
        1: "This person has a high blood pressure."},
    "HighChol": {0: "This person has normal cholesterol.",
        1: "This person has high cholesterol."},
    "CholCheck": {0: "This person has no cholesterol check in 5 years.",
        1: "This person has cholesterol checks in 5 years."},
    "BMI": "This person's Body Mass Index is {}",
    "Smoker": {0: "This person smoked less than 100 cigarettes in the
        entire life.",
        1: "This person smoked at least 100 cigarettes in the entire life
            ."},
    "Stroke": {0: "This person does not have a stroke.",
        1: "This person has a stroke."},
    "HeartDiseaseorAttack": {0: "This person does not have coronary heart
        disease (CHD) or myocardial infarction.",
        1: "This person has a coronary heart disease (CHD) or myocardial
            infarction."},
    "PhysActivity": {0: "This person did not have physical activities in
        the past 30 days.",
        1: "This person had physical activities in the past 30 days."},
    "Fruits": {0: "This person does not consume fruit every day.",
        1: "This person consumes fruit one or more times every day."},
    "Veggies": {0: "This person does not consume vegetables every day.",
        1: "This person consumes vegetables one or more times every day."
            },
    "HvyAlcoholConsump": {0: "This person is not a heavy drinker (adult
        men having more than 14 drinks per week and adult women having
        more than 7 drinks per week).",
        1: "This person is a heavy drinker (adult men having more than 14
            drinks per week and adult women having more than 7 drinks
            per week)."},
    "AnyHealthcare": {0: "This person does not Have any kind of health
        care coverage, including health insurance, prepaid plans such as
        HMO.",
        1: "This person has any kind of health care coverage, including
            health insurance, prepaid plans such as HMO."},
    "NoDocbcCost": {0: "This person never misses a doctor because of cost
        in the past 12 months.",
        1: "This person once needed to see a doctor but could not because
            of cost in the past 12 months."},
    "GenHlth": "This person's general health score is {} (1 represents
        the best, and 5 represents the worst).",
    "MentHlth": "This person had stress, depression, or problems with
        emotions in {} days of the past 30 days.",
    "PhysHlth": "This person had a physical illness or injury in {} days
        of the past 30 days.",
    "DiffWalk":  {0: "This person does not have serious difficulty
        walking or climbing stairs.",
        1: "This person has serious difficulty walking or climbing stairs
            ."},
    "Sex":  {0: "This person is a female.",
        1: "This person is a male."},
    "Age": "This person's age is {}.",
    "Education":  {
        1: "This person never attended school or only kindergarten.",
        2: "This person has grades 1 through 8 (Elementary).",
        3: "This person has grades 9 through 11 (Some high school).",
        4: "This person has grade 12 or GED (High school graduate).",
        5: "This person has college 1 year to 3 years (Some college or
            technical school).",
        6: "This person has college 4 years or more (College graduate).",
    },
    ...
```

Figure 9: Table to Text data template on the Diabete dataset (i).

```
1      ...
2      "Income":  {
3          1: "This person's income is less than 10000 dollars.",
4          2: "This person's income is more than 10000 dollars but less than
              15000 dollars.",
5          3: "This person's income is more than 15000 dollars but less than
              20000 dollars.",
6          4: "This person's income is more than 20000 dollars but less than
              25000 dollars.",
7          5: "This person's income is more than 25000 dollars but less than
              35000 dollars.",
8          6: "This person's income is more than 35000 dollars but less than
              55000 dollars.",
9          7: "This person's income is more than 55000 dollars but less than
              75000 dollars.",
10         8: "This person's income is more than 75000 dollars.",
11     },
12 }
```

Figure 10: Table to Text data template on the Diabete dataset (ii).

