# OpenReview forum: "Advancing Table Understanding of Large Language Models via Feature Re-ordering"
_ICLR.cc/2025/Conference — ICLR 2025 Conference Withdrawn Submission_

### Official Review · Reviewer_6LED · 2024-10-27

**Soundness:** 3
**Presentation:** 3
**Contribution:** 3
**Rating:** 6
**Confidence:** 4

**Summary:**

The paper explores the order bias in LLMs when handling tabular data. The authors propose ROTATOR-LLM, a method that dynamically reorders tabular features during inference, optimizing their sequence to enhance LLM performance without requiring model fine-tuning. This approach frames the ordering as a feature trajectory problem, solved using dynamic programming and a meta-controller. Experiments demonstrate that ROTATOR-LLM achieves notable improvements in accuracy across multiple datasets and LLMs.

**Strengths:**

1. **Soundness of approach.** The dynamic programming method for learning the value function is intuitive and straightforward to implement. Also, being able to use a relatively small model as the meta-controller makes the approach potentially more broadly applicable.

2. **Performance.** The method demonstrates strong results, even when compared to state-of-the-art foundational tabular LLMs across various datasets.

3. **Experiments and analyses.** The paper presents a wide range of empirical results and analyses, including attention map analysis, transferability of the learned controller, and comparisons with foundational tabular LLMs.

**Weaknesses:**

1. **Limited diversity of datasets.** Although the method is evaluated on four datasets, expanding to more diverse and complex datasets would strengthen the empirical results. Additionally, exploring whether a similar order bias exists for regression tasks and whether the approach can be applied to those cases would further enhance the findings.

2. **Scalability to larger datasets.** The dynamic programming approach may potentially struggle with a larger number of features, requiring much more trajectory data to train the meta-controller. A more thorough discussion and empirical evaluation of the method's scalability would be beneficial.

**Questions:**

1. How does the method perform on significantly larger datasets or highly imbalanced data?

2. Does the ordering learned by the meta-controller have anything to do with the feature importance?

3. How do controllers learned for different models compare in terms of the orders they predict?

---

### Official Review · Reviewer_e9wU · 2024-10-30

**Soundness:** 2
**Presentation:** 3
**Contribution:** 2
**Rating:** 5
**Confidence:** 3

**Summary:**

The paper presents a method for LLMs to handle tabular data better by reordering the table fields. LLMs are sensitive to the ordering of the context data. The proposed method leverages a lightweight LLM model to rank fields before feeding them to the LLMs. The light-weight LLM is trained using the classification loss of the base LLM based on permutation of the input fields. The approach is evaluated against several benchmarks and shows boosted performance. The authors also showed that the help from the ranking LLM could transfer across tasks.

**Strengths:**

1. The authors touched an interesting issue in tabular data processing where the table fields are random. Such randomness could cause big performance fluctuation in LLMs.
2. The presented method showed consistent performance improvements across benchmarks.

**Weaknesses:**

1. Not enough ablation studies to debunk what really helps on performance. There are a few confounders which could be nice to be addressed. The proposed approach uses the ranking LLM like inference time ranking or model ensemble. When ranking the input fields, the base LLM model essentially takes as input ranking LLM classification results. Are we able to tell this improvement is from the input fields ordering or it's just because the model ensemble effect.
2. The most challenging part of the proposed method would be that whether the ranking LLM's help can be transferred across different tasks. The authors only investigated the transferablility of the ranking LLM across different models.

**Questions:**

1. Would that make more sense if we train the ranking LLM using the binary classification rather than base model classification loss since base model eval could take a lot more compute?

---

### Official Review · Reviewer_hhrj · 2024-10-30

**Soundness:** 2
**Presentation:** 3
**Contribution:** 2
**Rating:** 3
**Confidence:** 3

**Summary:**

The paper addresses the challenge of enabling Large Language Models (LLMs) to better understand and process tabular data. LLMs, primarily trained on sequential natural language, struggle with tabular data because tables lack an intrinsic order of features (table fields). When tabular data is serialized for input into LLMs, an artificial order is imposed, which can significantly impact the model's performance on tasks related to tabular data understanding.

To mitigate this issue, the authors propose ROTATOR-LLM (Re-Ordering Tabular feATures fOR LLM), a simple and cost-effective method that optimizes the feature order of tabular data at test time without fine-tuning the base LLM. ROTATOR-LLM reframes the feature ordering problem as a feature trajectory generation task. It employs a dynamic programming-based meta-controller that auto-regressively generates an individualized feature trajectory for each data instance. This is achieved by estimating the accumulative value of the serialized feature input based on the LLM's final performance metrics and iteratively selecting features to maximize model performance.

Experimental results on multiple datasets and LLMs demonstrate that ROTATOR-LLM achieves significant performance boosts—close to or over 20% improvement compared to the un-ordered counterparts. It also outperforms state-of-the-art tabular LLM methods by a significant margin. Additionally, the meta-controller exhibits strong transferability across different LLMs, indicating that a controller trained on one model can enhance performance when applied to another.

**Strengths:**

## Originality
### Novel Insight into Feature Ordering
The paper focuses on feature ordering in tabular data when processed by LLMs. It brings attention to how the artificial order imposed during serialization can significantly influence model performance. The study may be valuable to better understand how sequential order of input impacts the LLMs output.

### Innovative Methodology
By reframing the feature ordering problem as a feature trajectory generation task and using a dynamic programming-based meta-controller, the authors introduce a creative approach to optimize feature order without modifying the base LLM.

## Quality
### Performance Improvements
The proposed method demonstrates substantial performance gains (close to or over 20%) over default or random feature ordering. It also outperforms existing state-of-the-art tabular LLM methods.

## Clarity (positives)
### Clear Presentation
The paper is well-organized, with logical flow across sections that detail the motivation, methodology, experiments, and results. Most of the concepts is communicated clearly. Single paragraphs and sentences reads well, without the need for rereading.

## Significance
### Cost-Effective Solution
ROTATOR-LLM operates at test time without requiring fine-tuning of the base LLM, which may save computations.

### Transferability of the Meta-Controller
The meta-controller trained on one LLM shows good transferability to other LLMs, indicating that the learned feature ordering strategy is robust and generalizable across models.

**Weaknesses:**

I have four major issues that stop me from giving it a higher note.

## 1. Fundamental Conceptual Issues with Feature Reordering

There is an inherent flaw in the approach of seeking an optimal feature order across the dataset instead of designing models capable of processing tabular data without such manipulations. Relying on reordering features to mitigate the ordering bias addresses a symptom rather than the root cause—the LLM's sensitivity to the sequential input of features. Ideally, models should be robust to feature order, especially given that in tabular data, the arrangement of columns is generally arbitrary and does not convey additional meaning. Furthermore, if the optimal feature order differs for each individual data point, the meta-controller may not converge to a single consistent order when trained on different subsets of the data. For instance, dividing the dataset into ten parts could lead to the meta-controller identifying different optimal orders for each subset, indicating that there is no universal optimal order. This variability raises questions about the practicality and scalability of the method, as it may not be feasible to train and maintain multiple controllers for different data partitions. It also suggests that the performance gains may not generalize well across diverse datasets or in real-world applications where data characteristics can vary significantly.

## 2. Weak Baselines and Comparisons

The baselines used in the experiments may not be strong enough to convincingly demonstrate the superiority of ROTATOR-LLM. Comparing against more robust and diverse baselines would provide a better assessment of the method's effectiveness. Some of the baselines that should be considered include:

1. Prompting the LLM to Reorder Features: Instead of using a fixed feature order, prompt the LLM to reorder the features itself and use that order. This could involve experimenting with different prompts or formatting examples to leverage the LLM's few-shot in-context learning abilities. By doing so, the LLM's inherent capabilities can be utilized without additional complexity.

2. Multiple Random Orders: Evaluating the model's performance using multiple random feature orders, possibly duplicating the information, to assess the impact of feature order variability without the need for a complex controller. This baseline would help determine if the benefits of feature reordering are due to the specific order found by the meta-controller or can be simply achieved by exposing the model to diverse orders.

3. Feature Importance-Based Ordering: Ordering features based on their importance as determined by traditional machine learning models (e.g., Random Forest, XGBoost). This could provide a simple yet effective baseline for feature ordering, leveraging established techniques in feature selection and importance weighting.

4. Iterative Feature Selection: The paper demonstrates that utilizing 10 features instead of 16 can yield improved performance on certain datasets. It would be valuable to assess whether recursive feature elimination alone is adequate in these instances. Alternatively, the authors could employ a greedy feature addition strategy, sequentially incorporating the features that provide the most significant performance enhancements first.

5. Comparison with Traditional Models: Since the task involves classification or regression over tabular data, it may be helpful to establish a baseline based on classical machine learning models like Random Forests or XGBoost. This would help contextualize the gains achieved by the proposed method and LLMs.

By not including these stronger baselines, the paper misses an opportunity to demonstrate that ROTATOR-LLM offers substantial improvements over simpler or more traditional approaches. Thus, forcing me to question the necessity of introducing the complex solution.
It is unclear whether such a complex method is required to address the feature ordering bias. Simpler solutions, such as heuristic-based ordering strategies might provide comparable improvements without the added computational overhead. The paper does not provide sufficient justification for choosing a meta-controller over these simpler alternatives.


## 3. Limited Discussion of Limitations
The paper does not explore the limitations of the proposed method. For instance, it does not discuss scenarios where feature reordering might not lead to performance gains or could potentially degrade performance. How robust is the method across a spectrum of datasets remains unclear as the paper only validates it on a handful of datasets.
Also, regarding transferability to other datasets: While the meta-controller shows some transferability across LLMs, it's unclear whether the approach is transferable across different datasets, especially those with different feature characteristics or domains.


## 4. Clarity and Presentation Issues
1. Writing Style: There are instances of awkward phrasing and grammatical errors. For example, sentences like "We demonstrate novelly discover and thoroughly explore the order bias..." contain mistakes that should be corrected. It is very minor in this paper, but rereading it should help. Above that, the citation style is a bit unfortunate. Please put the citations in between brackets.

2. Claims of Optimality and Maximization: The use of terms like "optimally" (e.g., line #370) and "maximally leverages" (e.g., line #477) are overstatements without rigorous proof or exhaustive empirical evidence. There is no indication that all possible permutations were tried to prove optimality and such claims should be moderated unless substantiated.

3. The methodology does not adequately address why duplication occurs in the reordered features, whereas the experiments do not show how it impacts model performance.

4. The paper overemphasizes attention entropy as a measure of attention concentration, but it is not clearly established how this directly correlates with performance improvements. Additional analysis or justification connecting attention patterns to model accuracy would strengthen this argument.

5. A visualization of how the meta controller generates feature trajectory would be helpful for a reader and lower the barrier to understanding the contribution.

**Questions:**

See Weaknesses section

---

### Official Review · Reviewer_xTwE · 2024-11-03

**Soundness:** 3
**Presentation:** 2
**Contribution:** 2
**Rating:** 5
**Confidence:** 3

**Summary:**

The paper proposes a new method to solve the positional bias issues in LLMs when solving table understanding. The method reframes feature ordering as a trajectory generation task though a meta-controller. Specifically, they adopt dynamic programming to create custom feature orderings for each data instance efficiently.

**Strengths:**

1. It is indeed an important point for LLMs to consider the order of input features, which may highly influence the performance.
2. The proposed method conducts test-time computing without fine-tuning the base LLM. This is very friendly to the real practice.
3. The performance boost is very remarkable. With the newly proposed method, the performance on various datasets improves by about 20%, which further demonstrates that feature ordering is important to LLMs' performance.

**Weaknesses:**

1. The problem setting is not very clear. The paper focuses on the table understanding while the so-called "tabular feature" is not introduced clearly. Existing work may represent a table in markdown/JSON/XML formats to LLMs while the "feature" needs further definition. Are the features the cells in the table? Please explain the problem settings more clearly.

2. The baseline choices are too limited. In the paper, the proposed method is only compared with Default/Random. However, the meta-controller is trained on the training set while these baselines don't learn anything from the training data. The comparison is not fair toward the simple baselines.

**Questions:**

I don't understand the motivation to convert a table into text features. Are there any explanations for this problem setting?

---

### Note · Authors · 2024-12-18

I have read and agree with the venue's withdrawal policy on behalf of myself and my co-authors.